# ONLINE STRUCTURE LEARNING FOR SUM-PRODUCT NETWORKS WITH GAUSSIAN LEAVES

**Wilson Hsu, Agastya Kalra & Pascal Poupart**
David R. Cheriton School of Computer Science
University of Waterloo
Waterloo, Ontario, Canada
{wwhsu,a6kalra,ppoupart}@uwaterloo.ca

## ABSTRACT

Sum-product networks have recently emerged as an attractive representation due to their dual view as a special type of deep neural network with clear semantics and a special type of probabilistic graphical model for which inference is always tractable. Those properties follow from some conditions (i.e., completeness and decomposability) that must be respected by the structure of the network. As a result, it is not easy to specify a valid sum-product network by hand and therefore structure learning techniques are typically used in practice. This paper describes the first *online* structure learning technique for continuous SPNs with Gaussian leaves. We also introduce an accompanying new parameter learning technique.

## 1 INTRODUCTION

Sum-product networks (SPNs) were first introduced by Poon & Domingos (2011) as a new type of deep representation. They distinguish themselves from other types of neural networks by several desirable properties:

1. The quantities computed by each node can be clearly interpreted as (un-normalized) probabilities.
2. SPNs are equivalent to Bayesian and Markov networks (Zhao et al., 2015) while ensuring that exact inference has linear complexity with respect to the size of the network.
3. They represent generative models that naturally handle arbitrary queries with missing data while changing which variables are treated as inputs and outputs.

There is a catch: these nice properties arise only when the structure of the network satisfies certain conditions (i.e., decomposability and completeness) (Poon & Domingos, 2011). Hence, it is not easy to specify sum-product networks by hand. In particular, fully connected networks typically violate those conditions. Similarly, most sparse structures that are handcrafted by practitioners to compute specific types of features or embeddings also violate those conditions. While this may seem like a major drawback, the benefit is that researchers have been forced to develop structure learning techniques to obtain valid SPNs that satisfy those conditions (Dennis & Ventura, 2012; Gens & Domingos, 2013; Peharz et al., 2013; Lee et al., 2013; Rooshenas & Lowd, 2014; Adel et al., 2015; Vergari et al., 2015; Rahman & Gogate, 2016; Mazen Melibari, 2016). At the moment, the search for good network structures in other types of neural networks is typically done by hand based on intuitions as well as trial and error. However the expectation is that automated structure learning techniques will eventually dominate. For this to happen, we need structure learning techniques that can scale easily to large amounts of data.

To that effect, we propose the first *online* structure learning technique for SPNs with Gaussian leaves. The approach starts with a network structure that assumes that all variables are independent. This network structure is then updated as a stream of data points is processed. Whenever a statistically significant correlation is detected between some variables, a correlation is introduced in the network in the form of a multivariate Gaussian or a mixture distribution. This is done while ensuring that the resulting network structure is necessarily valid. The approach is evaluated on several large benchmark datasets.

The paper is structured as follows. Section 2 provides some background about sum-product networks. Section 3 describes our *online* structure learning technique for SPNs with Gaussian leaves. Section 4 evaluates the performance of our structure learning technique on several large benchmark datasets. Finally, Section 5 concludes the paper and discusses possible directions for future work.

## 2 BACKGROUND

Sum-product networks (SPNs) were first proposed by Poon & Domingos (2011) as a new type of deep architecture consisting of a rooted acyclic directed graph with interior nodes that are sums and products while the leaves are tractable distributions, including Bernoulli distributions for discrete SPNs and Gaussian distributions for continuous SPNs. The edges emanating from sum nodes are labeled with non-negative weights $w$. An SPN encodes a function $f(\mathbf{X} = \mathbf{x})$ that takes as input a variable assignment $\mathbf{X} = \mathbf{x}$ and produces an output at its root. This function is defined recursively at each node $n$ as follows:

$$f_n(\mathbf{X} = \mathbf{x}) = \begin{cases} \Pr(\mathbf{X_n} = \mathbf{x_n}) & \text{if } isLeaf(n) \\ \sum_i w_i f_{child_i(n)}(\mathbf{x}) & \text{if } isSum(n) \\ \prod_i f_{child_i(n)}(\mathbf{x}) & \text{if } isProduct(n) \end{cases} \quad (1)$$

Here, $\mathbf{X_n} = \mathbf{x_n}$ denotes the variable assignment restricted to the variables contained in the leaf $n$. If none of the variables in leaf $n$ are instantiated by $\mathbf{X} = \mathbf{x}$ then $\Pr(\mathbf{X_n} = \mathbf{x_n}) = \Pr(\emptyset) = \mathbf{1}$. Note also that if leaf $n$ contains continuous variables, then $\Pr(\mathbf{X_n} = \mathbf{x_n})$ should be interpreted as $pdf(X_n = x_n)$.

An SPN is a neural network in the sense that each interior node can be interpreted as computing a linear combination of its children followed by a potentially non-linear activation function. Without loss of generality, assume that the SPN is organized in alternating layers of sums and product nodes.[1] It is easy to see that sum-nodes compute a linear combination of their children. Product nodes can be interpreted as the sum of its children in the log domain. Hence sum-product networks can be viewed as neural networks with logarithmic and exponential activation functions.

An SPN can also be viewed as encoding a joint distribution over the random variables in its leaves when the network structure satisfies certain conditions. These conditions are often defined in terms of the notion of *scope*.

**Definition 1** (Scope). *The $scope(n)$ of a node $n$ is the set of variables that are descendants of $n$.*

A sufficient set of conditions to ensure a valid joint distribution includes:

**Definition 2** (Completeness (Poon & Domingos, 2011)). *An SPN is complete if all children of the same sum node have the same scope.*

**Definition 3** (Decomposability (Poon & Domingos, 2011)). *An SPN is decomposable if all children of the same product node have disjoint scopes.*

Here decomposability allows us to interpret product nodes as computing factored distributions with respect to disjoint sets of variables, which ensures that the product is a valid distribution over the union of the scopes of the children. Similarly, completeness allows us to interpret sum nodes as computing a mixture of the distributions encoded by the children since they all have the same scope. Each child is a mixture component with mixture probability proportional to its weight. Hence, in complete and decomposable SPNs, the sub-SPN rooted at each node can be interpreted as encoding an (un-normalized) joint distribution over its scope. We can use the function $f$ to answer inference queries with respect to the joint distribution encoded by the entire SPN as follows:

- Marginal queries: $\Pr(\mathbf{X} = \mathbf{x}) = \frac{\mathbf{f_{root}(X=x)}}{\mathbf{f_{root}(\emptyset)}}$

- Conditional queries: $\Pr(\mathbf{X} = \mathbf{x} | \mathbf{Y} = \mathbf{y}) = \frac{\mathbf{f_{root}(X=x,Y=y)}}{\mathbf{f_{root}(Y=y)}}$

Unlike most neural networks that can answer only queries with fixed inputs and outputs, SPNs can answer conditional inference queries with varying inputs and outputs simply by changing the set of

---

[1]Consecutive sum nodes can always be merged into a single sum node. Similarly, consecutive product nodes can always be merged into a single product node.

variables that are queried (outputs) and conditioned on (inputs). Furthermore, SPNs can be used to generate data by sampling from the joint distributions they encode. This is achieved by a top-down pass through the network. Starting at the root, each child of a product node is followed, a single child of a sum node is sampled according to the unnormalized distribution encoded by the weights of the sum node and a variable assignment is sampled in each leaf that is reached. This is particularly useful in natural language generation tasks and image completion tasks (Poon & Domingos, 2011).

Note also that inference queries can be answered exactly in linear time with respect to the size of the network since each query requires two evaluations of the network function $f$ and each evaluation is performed in a bottom-up pass through the network. This means that SPNs can also be viewed as a special type of tractable probabilistic graphical model, in contrast to Bayesian and Markov networks for which inference is #P-hard (Roth, 1996). Any SPN can be converted into an equivalent bipartite Bayesian network without any exponential blow up, while Bayesian and Markov networks can be converted into equivalent SPNs at the risk of an exponential blow up (Zhao et al., 2015).

## 2.1 PARAMETER LEARNING

The weights of an SPN are its parameters. They can be estimated by maximizing the likelihood of a dataset (generative training) (Poon & Domingos, 2011) or the conditional likelihood of some output features given some input features (discriminative training) by Stochastic Gradient Descent (SGD) (Gens & Domingos, 2012). Since SPNs are generative probabilistic models where the sum nodes can be interpreted as hidden variables that induce a mixture, the parameters can also be estimated by Expectation Maximization (EM) (Poon & Domingos, 2011; Peharz, 2015). Zhao & Poupart (2016) provides a unifying framework that explains how likelihood maximization in SPNs corresponds to a signomial optimization problem where SGD is a first order procedure, one can also consider a sequential monomial approximation and EM corresponds to a concave-convex procedure that converges faster than the other techniques. Since SPNs are deep architectures, SGD and EM suffer from vanishing updates and therefore "hard" variants have been proposed to remedy to this problem (Poon & Domingos, 2011; Gens & Domingos, 2012). By replacing all sum nodes by max nodes in an SPN, we obtain a max-product network where the gradient is constant (hard SGD) and latent variables become deterministic (hard EM). It is also possible to train SPNs in an online fashion based on streaming data (Lee et al., 2013; Rashwan et al., 2016; Zhao et al., 2016; Jaini et al., 2016). In particular, it was shown that online Bayesian moment matching (Rashwan et al., 2016; Jaini et al., 2016) and online collapsed variational Bayes (Zhao et al., 2016) perform much better than SGD and online EM.

## 2.2 STRUCTURE LEARNING

Since it is difficult to specify network structures for SPNs that satisfy the decomposability and completeness properties, several automated structure learning techniques have been proposed (Dennis & Ventura, 2012; Gens & Domingos, 2013; Peharz et al., 2013; Lee et al., 2013; Rooshenas & Lowd, 2014; Adel et al., 2015; Vergari et al., 2015; Rahman & Gogate, 2016; Mazen Melibari, 2016). The first two structure learning techniques (Dennis & Ventura, 2012; Gens & Domingos, 2013) are top down approaches that alternate between instance clustering to construct sum nodes and variable partitioning to construct product nodes. We can also combine instance clustering and variable partitioning in one step with a rank-one submatrix extraction by performing a singular value decomposition (Adel et al., 2015). Alternatively, we can learn the structure of SPNs in a bottom-up fashion by incrementally clustering correlated variables (Peharz et al., 2013). These algorithms all learn SPNs with a tree structure and univariate leaves. It is possible to learn SPNs with multivariate leaves by using a hybrid technique that learns an SPN in a top down fashion, but stops early and constructs multivariate leaves by fitting a tractable probabilistic graphical model over the variables in each leaf (Rooshenas & Lowd, 2014; Vergari et al., 2015). It is also possible to merge similar subtrees into directed acyclic graphs in a post-processing step to reduce the size of the resulting SPN (Rahman & Gogate, 2016). Furthermore, Mazen Melibari (2016) proposed dynamic SPNs for variable length data and described a search-and-score structure learning technique that does a local search over the space of network structures.

So far, all these structure learning algorithms are batch techniques that assume that the full dataset is available and can be scanned multiple times. Lee et al. (2013) describes an online structure

learning technique that gradually grows a network structure based on mini-batches. The algorithm is a variant of LearnSPN (Gens & Domingos, 2013) where the clustering step is modified to use online clustering. As a result, sum nodes can be extended with more children when the algorithm encounters a mini-batch that is better clustered with additional clusters. Product nodes are never modified after their creation.

Since existing structure learning techniques have all been designed for discrete SPNs and have yet to be extended to continuous SPNs such as Gaussian SPNs, the state of the art for continuous (and large scale) datasets is to generate a random network structure that satisfies decomposability and completeness after which the weights are learned by a scalable online learning technique (Jaini et al., 2016). We advance the state of the art by proposing a first online structure learning technique for Gaussian SPNs.

# 3 PROPOSED ALGORITHM

In this work, we assume that the leaf nodes all have Gaussian distributions. A leaf node may have more than one variable in its scope, in which case it follows a multivariate Gaussian distribution.

Suppose we want to model a probability distribution over a $d$-dimensional space. The algorithm starts with a fully factorized joint probability distribution over all variables, $p(\mathbf{x}) = p(x_1, x_2, \ldots, x_d) = p_1(x_1)p_2(x_2)\cdots p_d(x_d)$. This distribution is represented by a product node with $d$ children, the $i$th of which is a univariate distribution over the variable $x_i$. Therefore, initially we assume that the variables are independent, and the algorithm will update this probability distribution as new data points are processed.

Given a mini-batch of data points, the algorithm passes the points through the network from the root to the leaf nodes and updates each node along the way. This update includes two parts:

- updating the parameters of the SPN, and
- updating the structure of the network.

## 3.1 PARAMETER UPDATE

The parameters are updated by keeping track of running sufficient statistics. There are two types of parameters in the model: weights on the branches under a sum node, and parameters for the Gaussian distribution in a leaf node.

We propose a new online algorithm for parameter learning that is simple while ensuring that after each update, the likelihood of the last processed data point is increased (similar to stochastic gradient ascent). Algorithm 1 describes the pseudocode of this procedure. Every node in the network has a count, $n_c$, initialized to 1. When a data point is received, the likelihood of this data point is computed at each node. Then the parameters of the network are updated in a recursive top-down fashion by starting at the root node. When a sum node is traversed, its count is increased by 1 and the count of the child with the highest likelihood is increased by 1. This effectively increases the weight of the child with the highest likelihood while decreasing the weights of the remaining children. As a result, the overall likelihood at the sum node will increase. The weight $w_{s,c}$ of a branch between a sum node $s$ and one of its children $c$ can then be estimated as

$$w_{s,c} = \frac{n_c}{n_s} \qquad (2)$$

where $n_s$ is the count of the sum node and $n_c$ is the count of the child node. We also recursively update the subtree of the child with the highest likelihood. In the case of ties, we simply choose one of the children with highest likelihood at random to be updated.

Since there are no parameters associated with a product node, the only way to increase its likelihood is to increase the likelihood at each of its children. We increment the count at each child of a product node and recursively update the subtrees rooted at each child.

Since each leaf node represents a Gaussian distribution, it keeps track of the empirical mean vector $\mu$ and empirical covariance matrix $\Sigma$ for the variables in its scope. When a leaf node with a current

---
**Algorithm 1** parameterUpdate(root(SPN),data)

---
**Input:** SPN and $m$ data points
**Output:** SPN with updated parameters
 $n_{root} \leftarrow n_{root} + m$
 **if** $isProduct(root)$ **then**
 **for** each $child$ of $root$ **do**
 $parameterUpdate(child, data)$
 **end for**
 **else if** $isSum(root)$ **then**
 **for** each $child$ of $root$ **do**
 $subset \leftarrow \{x \in data \mid likelihood(child, x) \geq likelihood(child', x) \; \forall child' \text{ of } root\}$
 $parameterUpdate(child, subset)$
 $w_{root,child} \leftarrow \frac{n_{child}+1}{n_{root}+\#children}$
 **end for**
 **else if** $isLeaf(root)$ **then**
 update mean $\mu^{(root)}$ based on Eq. 3
 update covariance matrix $\Sigma^{(root)}$ based on Eq. 4
 **end if**

---

count of $n$ receives a batch of $m$ data points $x^{(1)}, x^{(2)}, \ldots, x^{(m)}$, the empirical mean and empirical covariance are updated according to the equations:

$$\mu_i' = \frac{1}{n+m} \left( n\mu_i + \sum_{k=1}^{m} x_i^{(k)} \right) \tag{3}$$

and

$$\Sigma_{i,j}' = \frac{1}{n+m} \left[ n\Sigma_{i,j} + \sum_{k=1}^{m} \left( x_i^{(k)} - \mu_i \right) \left( x_j^{(k)} - \mu_j \right) \right] - (\mu_i' - \mu_i)(\mu_j' - \mu_j) \tag{4}$$

where $i$ and $j$ index the variables in the leaf node's scope, and $\mu'$ and $\Sigma'$ are the new mean and covariance after the update.

This parameter update technique is related to, but different from hard SGD and hard EM used in (Poon & Domingos, 2011; Gens & Domingos, 2012; Lee et al., 2013). Hard SGD and hard EM also keep track of a count for the child of each sum node and increment those counts each time a data point reaches this child. However, to decide when a child is reached by a data point, they replace all descendant sum nodes by max nodes and evaluate the resulting max-product network. In contrast, we retain the descendant sum nodes and evaluate the original sum-product network as it is. This evaluates more faithfully the probability that a data point is generated by a child.

Alg. 1 does a single pass through the data. The complexity of updating the parameters after each data point is linear in the size of the network (i.e., # of edges) since it takes one bottom up pass to compute the likelihood of the data point at each node and one top-down pass to update the sufficient statistics and the weights. The update of the sufficient statistics can be seen as locally maximizing the likelihood of the data. The empirical mean and covariance of the Gaussian leaves locally increase the likelihood of the data that reach that leaf. Similarly, the count ratios used to set the weights under a sum node locally increase the likelihood of the data that reach each child. We prove this result below.

**Theorem 1.** *Let $\theta_s$ be the set of parameters of an SPN $s$, and let $f_s(\cdot|\theta_s)$ be the probability density function of the SPN. Given an observation $x$, suppose the parameters are updated to $\theta_s'$ based on the running average update procedure, then we have $f_s(x|\theta_s') \geq f_s(x|\theta_s)$.*

*Proof.* We will prove the theorem by induction. First suppose the SPN is just one leaf node. In this case, the parameters are the empirical mean and covariance, which is the maximum likelihood estimator for Gaussian distribution. Suppose $\theta$ consists of the parameters learned using $n$ data points $x^{(1)}, \ldots, x^{(n)}$, and $\theta'$ consists of the parameters learned using the same $n$ data points and an

additional observation $x$. Then we have

$$f_s(x|\theta'_s) \prod_{i=1}^{n} f(x^{(i)}|\theta'_s) \geq f_s(x|\theta_s) \prod_{i=1}^{n} f_s(x^{(i)}|\theta_s) \geq f_s(x|\theta_s) \prod_{i=1}^{n} f_s(x^{(i)}|\theta'_s) \tag{5}$$

Thus we get $f_s(x|\theta'_s) \geq f_s(x|\theta_s)$.

Now suppose we have an SPN $s$ where each child SPN $t$ satisfies the property $f_t(x|\theta'_t) \geq f_t(x|\theta_t)$. If the root of $s$ is a product node, then $f_s(x|\theta'_s) = \prod_t f_t(x|\theta'_t) \geq \prod_t f_t(x|\theta_t) = f_s(x|\theta_s)$.

Now suppose the root of $s$ is a sum node. Let $n_t$ be the count of child $t$, and let $u = \arg\max_t f_t(x|\theta_t)$. Then we have

$$
\begin{aligned}
f_s(x|\theta'_s) &= \frac{1}{n+1}\left( f_u(x|\theta'_u) + \sum_t n_t f_t(x|\theta'_t) \right) \\
&\geq \frac{1}{n+1}\left( f_u(x|\theta_u) + \sum_t n_t f_t(x|\theta_t) \right) \quad \text{by inductive hypothesis} \\
&\geq \frac{1}{n+1}\left( \sum_t \frac{n_t}{n} f_t(x|\theta_t) + \sum_t n_t f_t(x|\theta_t) \right) \\
&= \frac{1}{n}\sum_t n_t f_t(x|\theta_t) \\
&= f_s(x|\theta_s) \quad \square
\end{aligned}
$$

## 3.2 STRUCTURE UPDATE

The simple online parameter learning described above can be easily extended to enable online structure learning. Algorithm 2 describes the pseudocode of the resulting procedure called oSLRAU (online Structure Learning with Running Average Update). Similar to leaf nodes, each product node also keeps track of the empirical mean vector and empirical covariance matrix of the variables in its scope. These are updated in the same way as the leaf nodes.

Initially, when a product node is created, all variables in the scope are assumed independent (see Algorithm 5). As new data points arrive at a product node, the covariance matrix is updated, and if the absolute value of the Pearson correlation coefficient between two variables are above a certain threshold, the algorithm updates the structure so that the two variables become correlated in the model.

We correlate two variables in the model by combining the child nodes whose scopes contain the two variables. The algorithm employs two approaches to combine the two child nodes:

- create a multivariate leaf node (Algorithm 4), or
- create a mixture of two components over the variables (Algorithm 3).

These two processes are depicted in Figure 1. On the left, a product node with scope $x_1, \ldots, x_5$ originally has three children. The product node keeps track of the empirical mean and empirical covariance for these five variables. Suppose it receives a mini-batch of data and updates the statistics. As a result of this update, $x_1$ and $x_3$ now have a correlation above the threshold.

Figure 1 illustrates the two approaches to model this correlation. In the middle of Figure 1, the algorithm combines the two child nodes that have $x_1$ and $x_3$ in their scope, and turns them into a multivariate leaf node. Since the product node already keeps track of the mean and covariance of these variables, we can simply use those statistics as the parameters for the new leaf node.

Another way to correlate $x_1$ and $x_3$ is to create a mixture, as shown in the right part of Figure 1. The mixture has two components. The first component contains the original children of the product node that contain $x_1$ and $x_3$. The second component is a new product node, which is again initialized to have a fully factorized distribution over its scope (Alg. 5). The mini-batch of data points are then passed down the new mixture to update its parameters.

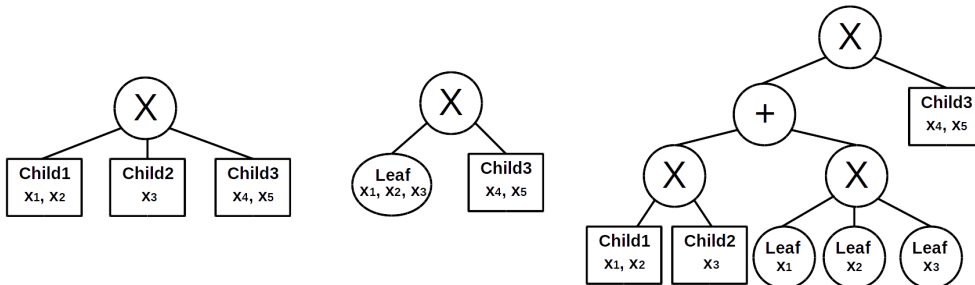

Figure 1: Depiction of how correlations between variables are introduced in the model. Left: original product node with three children. Middle: combine Child1 and Child2 into a multivariate leaf node (Alg. 4). Right: create a mixture to model the correlation (Alg. 3).

Note that although the children are drawn like leaf nodes in the diagrams, they can in fact be entire subtrees. Since the process does not involve the parameters in a child, it works the same way if some of the children are trees instead of single nodes.

The technique chosen to induce a correlation depends on the number of variables in the scope. The algorithm creates a multivariate leaf node when the combined scope of the two child nodes has a number of variables that does not exceed some threshold and if the total number of variables in the problem is greater than this threshold, otherwise it creates a mixture. Since the number of parameters in multivariate Gaussian leaves grows at a quadratic rate with respect to the number of variables, it is not advised to consider multivariate leaves with too many variables. In contrast, the mixture construction increases the number of parameters at a linear rate, which is less prone to overfitting when many variables are correlated.

To simplify the structure, if a product node ends up with only one child, it is removed from the network, and its only child is joined with its parent. Similarly, if a sum node ends up being a child of another sum node, then the child sum node can be removed, and all its children are promoted one layer up.

Note that the this structure learning technique does a single pass through the data and therefore is entirely online. The time and space complexity of updating the structure after each data point is linear in the size of the network (i.e., # of edges) and quadratic in the number of features (since product nodes store a covariance matrix that is quadratic in the size of their scope). The algorithm also ensures that the decomposability and completeness properties are preserved after each update.

Our algorithm (oSLRAU) is related to, but different from the online structure learning technique proposed by Lee et al. (2013). Lee et al.'s technique was applied to discrete datasets while oSLRAU learns SPNs with Gaussian leaves based on real-valued data. Furthermore, Lee et al.'s technique incrementally constructs a network in a top down fashion by adding children to sum nodes by online clustering. Once a product node is constructed, it is never modified. In contrast, oSLRAU incrementally constructs a network in a bottom up fashion by detecting correlations and modifying product nodes to represent these correlations. Finally, Lee et al.'s technique updates the parameters by hard EM (which implicitly works with a max-product network) while oSLRAU updates the parameters by Alg. 1 (which retains the original sum-product network) as explained in the previous section.

## 4 EXPERIMENTS

The source code for our new online structure learning algorithm is available at `github.com/whsu/spn`.

---

**Algorithm 2** $oSLRAU(root(SPN), data)$

---

**Input:** SPN and $m$ data points
**Output:** SPN with updated parameters

 $n_{root} \leftarrow n_{root} + m$
 **if** $isProduct(root)$ **then**
 update covariance matrix $\Sigma^{(root)}$ based on Eq. 4
 $highestCorrelation \leftarrow 0$
 **for** each $c, c' \in children(root)$ where $c \neq c'$ **do**

$$correlation_{c,c'} \leftarrow \max_{i \in scope(c), j \in scope(c')} \frac{|\Sigma_{ij}^{(root)}|}{\sqrt{\Sigma_{ii}^{(root)}\Sigma_{jj}^{(root)}}}$$

 **if** $correlation_{c,c'} > highestCorrelation$ **then**
 $highestCorrelation \leftarrow correlation_{c,c'}$
 $child_1 \leftarrow c$
 $child_2 \leftarrow c'$
 **end if**
 **end for**
 **if** $highest \geq threshold$ **then**
 **if** $|scope(child_1) \cup scope(child_2)| \geq nVars$ **then**
 $createMixture(root, child_1, child_2)$
 **else**
 $createMultivariateGaussian(root, child_1, child_2)$
 **end if**
 **end if**
 **for** each $child$ of $root$ **do**
 $oSLRAU(child, data)$
 **end for**
 **else if** $isSum(root)$ **then**
 **for** each $child$ of $root$ **do**
 $subset \leftarrow \{x \in data \mid likelihood(child, x) \geq likelihood(child', x) \; \forall child' \text{ of } root\}$
 $oSLRAU(child, subset)$
 $w_{root,child} \leftarrow \frac{n_{child}+1}{n_{root}+\#children}$
 **end for**
 **else if** $isLeaf(root)$ **then**
 update mean $\mu^{(root)}$ based on Eq. 3
 update covariance matrix $\Sigma^{(root)}$ based on Eq. 4
 **end if**

---

## 4.1 TOY DATASET

As a proof of concept, we first test the algorithm on a toy synthetic dataset. We generate data from the 3-dimensional distribution

$$p(x_1, x_2, x_3) = [0.25N(x_1|1, 1)N(x_2|2, 2) + 0.25N(x_1|11, 1)N(x_2|12, 2)$$
$$+ 0.25N(x_1|21, 1)N(x_2|22, 2) + 0.25N(x_1|31, 1)N(x_2|32, 2)]N(x_3|3, 3),$$

where $N(\cdot|\mu, \sigma^2)$ is the normal distribution with mean $\mu$ and variance $\sigma^2$.

Therefore, the first two dimensions $x_1$ and $x_2$ are generated from a Gaussian mixture with four components, and $x_3$ is independent from the other two variables.

Starting from a fully factorized distribution, we would expect $x_3$ to remain factorized after learning from data. Furthermore, the algorithm should generate new components along the first two dimensions as more data points are received since $x_1$ and $x_2$ are correlated.

This is indeed what happens. Figure 2 shows the structure learned after 200 and 500 data points. The variable $x_3$ remains factorized regardless of the number of data points seen, whereas more components are created for $x_1$ and $x_2$ as more data points are processed.

---

**Algorithm 3** $createMixture(root, child_1, child_2)$

---

**Input:** SPN and two children to be merged
**Output:** new mixture model
 remove $child_1$ and $child_2$ from $root$
 $component_1 \leftarrow$ create product node
 add $child_1$ and $child_2$ as children of $component_1$
 $n_{component_1} \leftarrow n_{root}$
 $jointScope \leftarrow scope(child_1) \cup scope(child_2)$
 $\Sigma^{(component_1)} \leftarrow \Sigma^{(root)}_{jointScope,jointScope}$
 $component_2 \leftarrow createFactoredModel(jointScope)$
 $n_{component_2} \leftarrow 0$
 $mixture \leftarrow$ create sum node
 add $component_1$ and $component_2$ as children of $mixture$
 $n_{mixture} \leftarrow n_{root}$
 $w_{mixture,component_1} \leftarrow \frac{n_{component_1}+1}{n_{mixture}+2}$
 $w_{mixture,component_2} \leftarrow \frac{n_{component_2}+1}{n_{mixture}+2}$
 add $mixture$ as child of $root$
 return $root$

---

**Algorithm 4** $createMultiVarGaussian(root, child_1, child_2)$

---

**Input:** SPN, two children to be merged and data
**Output:** new multivariate Gaussian
 create $multiVarGaussian$
 $jointScope \leftarrow \{scope(child_1) \cup scope(child_2)\}$
 $\mu^{(multiVarGaussian)} \leftarrow \mu^{(root)}_{jointScope}$
 $\Sigma^{(multiVarGaussian)} \leftarrow \Sigma^{(root)}_{jointScope,jointScope}$
 $n_{multiVarGaussian} \leftarrow n_{root}$
 return $multiVarGaussian$

---

**Algorithm 5** $createFactoredModel(scope)$

---

**Input:** scope (set of variables)
**Output:** fully factored SPN
 $factoredModel \leftarrow$ create product node
 **for** each $i \in scope$ **do**
 add $N_i(\mu=0, \sigma=\Sigma^{(root)}_{i,i})$ as child of $factoredModel$
 **end for**
 $\Sigma^{(factoredModel)} \leftarrow \mathbf{0}$
 $n_{factoredModel} \leftarrow 0$
 return $factoredModel$

---

Figure 3 shows the data points along the first two dimensions and the Gaussian components learned. We can see that the algorithm generates new components to model the correlation between $x_1$ and $x_2$ as it processes more data.

## 4.2 COMPARISON TO OTHER ALGORITHMS

In a second experiment, we compare our algorithm to several alternatives on the same datasets used by Jaini et al. (2016). We use 0.1 as the correlation threshold in all experiments, and we use mini-batch sizes of 1 for the three datasets with fewest instances (Quake, Banknote, Abalone), 8 for the two slightly larger ones (Kinematics, CA), and 256 for the two datasets with most instances (Flow Size, Sensorless).

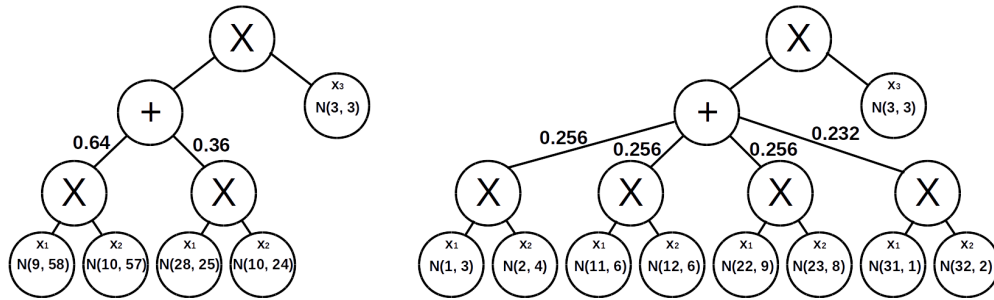

Figure 2: Learning the structure from the toy dataset using univariate leaf nodes. Left: after 200 data points. Right: after 500 data points.

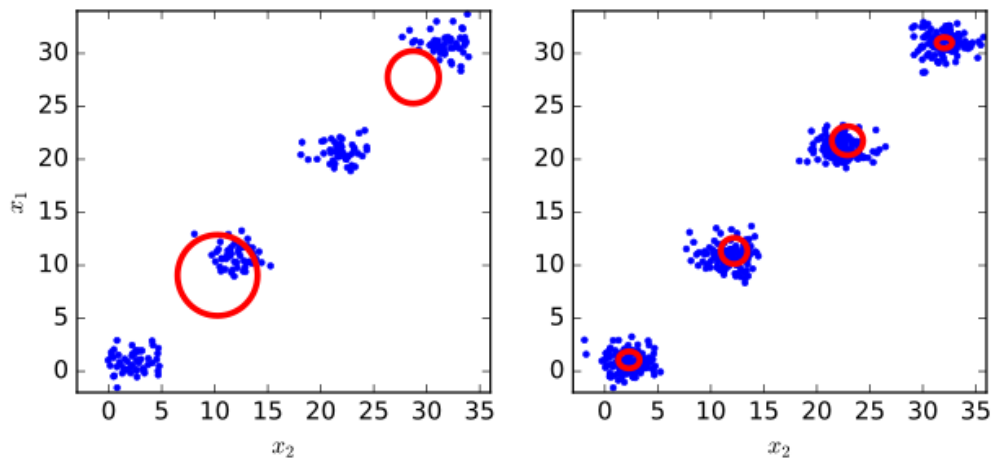

Figure 3: Blue dots are the data points from the toy dataset, and the red ellipses show the diagonal Gaussian components learned. Left: after 200 data points. Right: after 500 data points.

Table 1: Average log-likelihood scores with standard error on small real-world data sets. The best results are highlighted in bold. (random) indicates a random network structure and (GMM) indicates a fixed network structure corresponding to a Gaussian mixture model.

| Dataset<br># of vars | Flow Size<br>3 | Quake<br>4 | Banknote<br>4 | Abalone<br>8 | Kinematics<br>8 | CA<br>22 | Sensorless<br>48 |
|---|---|---|---|---|---|---|---|
| oSLRAU | **14.78**<br>± 0.97 | **-1.86**<br>± 0.20 | -2.04<br>± 0.15 | **-1.12**<br>± 0.21 | -11.15<br>± 0.03 | **17.10**<br>± 1.36 | **54.82**<br>± 1.67 |
| oBMM<br>(random) | - | - | - | -1.82<br>± 0.19 | -11.19<br>± 0.03 | -2.47<br>± 0.56 | 1.58<br>± 1.28 |
| oEM<br>(random) | - | - | - | -11.36<br>± 0.19 | -11.35<br>± 0.03 | -31.34<br>± 1.07 | -3.40<br>± 6.06 |
| oBMM<br>(GMM) | 4.80<br>± 0.67 | -3.84<br>± 0.16 | -4.81<br>± 0.13 | -1.21<br>± 0.36 | -11.24<br>± 0.04 | -1.78<br>± 0.59 | - |
| oEM<br>(GMM) | -0.49<br>± 3.29 | -5.50<br>± 0.41 | -4.81<br>± 0.13 | -3.53<br>± 1.68 | -11.35<br>± 0.03 | -21.39<br>± 1.58 | - |
| SRBM | -0.79<br>± 0.004 | -2.38<br>± 0.01 | -2.76<br>± 0.001 | -2.28<br>± 0.001 | **-5.55**<br>± 0.02 | -4.95<br>± 0.003 | -26.91<br>± 0.03 |
| GenMMN | 0.40<br>± 0.007 | -3.83<br>± 0.21 | **-1.70**<br>± 0.03 | -3.29<br>± 0.10 | -11.36<br>± 0.02 | -5.41<br>± 0.14 | -29.41<br>± 1.16 |

The experimental results for our algorithm called *online structure learning with running average update* (oSLRAU) are listed in Table 1 along with results reproduced from Jaini et al. (2016). The table reports the average test log likelihoods with standard error on 10-fold cross validation. oSLRAU achieved better log likelihoods than online Bayesian moment matching (oBMM) (Jaini et al., 2016) and online expectation maximization (oEM) (Cappé & Moulines, 2009) with network structures generated at random or corresponding to Gaussian mixture models (GMMs). This highlights the main advantage of oSLRAU: learning a structure that models the data. Stacked Restricted Boltzmann Machines (SRBMs) (Salakhutdinov & Hinton, 2009) and Generative Moment Matching Networks (GenMMNs) (Li et al., 2015) are other types of deep generative models. Since it is not possible to compute the likelihood of data points with GenMMNs, the model is augmented with Parzen windows. More specifically, 10,000 samples are generated using the resulting GenMMNs and a Gaussian kernel is estimated for each sample by adjusting its parameters to maximize the likelihood of a validation set. However, as pointed out by Theis et al. (2015) this method only provides an approximate estimate of the log-likelihood and therefore the log-likelihood reported for GenMMNs in Table 1 may not be directly comparable to the log-likelihood of other models.

The network structures for GenMMNs and SRBMs are fully connected while ensuring that the number of parameters is comparable to those of the SPNs. oSLRAU outperforms these models on 5 datasets while SRBMs and GenMMNs each outperform oSLRAU on one dataset. Although SRBMs and GenMMNs are more expressive than SPNs since they allow other types of nodes beyond sums and products, training GenMMNs and SRBMs is notoriously difficult. In contrast, oSLRAU provides a simple and effective way of optimizing the structure and parameters of SPNs that captures well the correlations between variables and therefore yields good results.

## 4.3 LARGE DATASETS

We also tested oSLRAU on larger datasets to evaluate its scaling properties. Table 2 shows the number of attributes and data points in each dataset. Table 3 compares the average log-likelihood of oSLRAU to that of randomly generated networks (which are the state of the art for obtain a valid continuous SPNs) for those large datasets. For a fair comparison we generated random networks that are at least as large as the networks obtained by oSLRAU. oSLRAU achieves higher log-likelihood than random networks since it effectively discovers empirical correlations and generates a structure that captures those correlations.

We also compare oSLRAU to a publicly available implementation of RealNVP[2]. Since the benchmarks include a variety of problems from different domains and it is not clear what network architecture would work best, we used a default 2-hidden-layer fully connected network. The two

---

[2]https://github.com/taesung89/real-nvp

Table 2: Information for each large dataset

| Dataset | Datapoints | Variables |
|---------|-----------|-----------|
| Voxforge | 3,603,643 | 39 |
| Power | 2,049,280 | 4 |
| Network | 434,873 | 3 |
| GasSen | 8,386,765 | 16 |
| MSD | 515,344 | 90 |
| GasSenH | 928,991 | 10 |

Table 3: Average log-likelihood scores with standard error on large real-world data sets. The best results among the online techniques (random, oSLRAU and RealNVP online) are highlighted in bold. Results for RealNVP offline are also included for comparison purposes.

| Datasets | Random | oSLRAU | RealNVP Online | RealNVP Offline |
|----------|--------|--------|----------------|-----------------|
| Voxforge | -33.9 ± 0.3 | **-29.6** ± 0.0 | -169.0 ± 0.6 | -168.2 ± 0.8 |
| Power | -2.83 ± 0.13 | **-2.46** ± 0.11 | -18.70 ± 0.19 | -17.85 ± 0.22 |
| Network | -5.34 ± 0.03 | **-4.27** ± 0.04 | -10.80 ± 0.02 | -7.89 ± 0.05 |
| GasSen | -114 ± 2 | **-102** ± 4 | -748 ± 99 | -443 ± 64 |
| MSD | -538.8 ± 0.7 | -531.4 ± 0.3 | **-362.4** ± 0.4 | -257.1 ± 2.03 |
| GasSenH | -21.5 ± 1.3 | **-15.6** ± 1.2 | -44.5 ± 0.1 | 44.2 ± 0.1 |

layers have the same size. For a fair comparison, we used a number of nodes per layer that yields approximately the same number of parameters as the sum product networks. Training was done by stochastic gradient descent in TensorFlow with a step size of 0.01 and mini-batch sizes that vary from 100 to 1500 depending on the size of the dataset. We report the results for online learning (single iteration) and offline learning (validation loss stops decreasing). In this experiment, the correlation threshold was kept constant at 0.1. To determine the maximum number of variables in multivariate leaves, we followed the following rule: at most one variable per leaf if the problem has 3 features or less and then increase the maximum number of variables per leaf up to 4 depending on the number of features. Further analysis on the effects of varying the maximum number of variables per leaf are available below. We do this to balance the size and the expressiveness of the resulting SPN. oSLRAU outperformed RealNVP on 5 of the 6 datasets. This can be explained by the fact that oSLRAU learns a structure that is suitable for each problem while RealNVP does not learn any structure. Note that it should be possible for RealNVP to obtain better results by using a better architecture than a default 2-hidden-layer network, however in the absence of domain knowledge this is difficult. Furthermore, in online learning with streaming data, it is not possible to do an offline search over some hyperparameters such as the number of layers and nodes in order to fine tune the architecture. Hence, the results presented in Table 3 highlight the importance of an online structure learning technique such as oSLRAU to obtain a suitable network structure with streaming data in the absence of domain knowledge.

Table 4 reports the training time (seconds) and the size (# of nodes) of the resulting SPNs for each dataset when running oSLRAU and a variant that stops structure learning early. The experiments were carried out on an Amazon c4.xlarge machine with 4 vCPUs (high frequency Intel Xeon E5-2666 v3 Haswell processors) and 7.5 Gb of RAM. The times are relatively short since oSLRAU is an online algorithm and therefore does a single pass through the data. Since it gradually constructs the structure of the SPN as it processes the data, we can also stop the updates to the structure early (while still updating the parameters). This helps to mitigate overfitting while producing much smaller SPNs and reducing the running time. In the columns labeled "early stop" we report the results achieved when structure learning is stopped after processing one ninth of the data. The resulting SPNs are significantly smaller, while achieving a log-likelihood that is close to that of oSLRAU without early stopping.

The size of the resulting SPNs and their log-likelihood also depend on the correlation threshold used to determine when the structure should be updated to account for a detected correlation, and the maximum size of a leaf node used to determine when to branch off into a new subtree.

Table 4: Large datasets: comparison of oSLRAU with and without early stopping (i.e., no structure learning after one ninth of the data is processed, but still updating the parameters).

| Dataset | log-likelihood | | time (sec) | | SPN size (# nodes) | |
|---|---|---|---|---|---|---|
| | oSLRAU | early stop | oSLRAU | early stop | oSLRAU | early stop |
| Power | $-2.46 \pm 0.11$ | $\mathbf{-0.24} \pm 0.20$ | 183 | 70 | 23360 | 1154 |
| Network | $\mathbf{-4.27} \pm 0.02$ | $-4.30 \pm 0.02$ | 14 | 4 | 7214 | 249 |
| GasSen | $\mathbf{-102} \pm 4$ | $-111 \pm 3$ | 351 | 188 | 5057 | 564 |
| MSD | $\mathbf{-531.4} \pm 0.3$ | $-534.9 \pm 0.3$ | 44 | 26 | 672 | 238 |
| GasSenH | $\mathbf{-15.6} \pm 1.2$ | $-18.6 \pm 1.0$ | 12 | 9 | 920 | 131 |

Table 5: Log likelihoods with standard error as we vary the threshold for the maximum # of variables in a multivariate Gaussian leaf. No results are reported (dashes) when the maximum # of variables is greater than the total number of variables.

| Dataset | Maximum # of Variables per Leaf Node | | | | |
|---|---|---|---|---|---|
| | 1 | 2 | 3 | 4 | 5 |
| Power | $\mathbf{-1.71} \pm 0.18$ | $-3.02 \pm 0.24$ | $-3.74 \pm 0.28$ | $-4.52 \pm 0.1$ | —— |
| Network | $\mathbf{-4.27} \pm 0.09$ | $-4.53 \pm 0.09$ | $-4.75 \pm 0.02$ | —— | —— |
| GasSen | $-105 \pm 2.5$ | $-103 \pm 2.8$ | $\mathbf{-102} \pm 4.1$ | $-104 \pm 3.8$ | $-103 \pm 3.8$ |
| MSD | $-532 \pm 0.32$ | $\mathbf{-531} \pm 0.32$ | $\mathbf{-531} \pm 0.28$ | $\mathbf{-531} \pm 0.31$ | $-532 \pm 0.34$ |
| GasSenH | $-17.2 \pm 1.04$ | $-16.8 \pm 1.23$ | $\mathbf{-15.6} \pm \mathbf{1.13}$ | $-15.9 \pm 1.3$ | $-16.1 \pm 1.47$ |

Table 6: Average times (seconds) as we vary the threshold for the maximum # of variables in a multivariate Gaussian leaf. No results are reported (dashes) when the maximum # of variables is greater than the total number of variables.

| Dataset | Maximum # of Variables per Leaf Node | | | | |
|---|---|---|---|---|---|
| | 1 | 2 | 3 | 4 | 5 |
| Power | 133 | 41.5 | 13.8 | 9.9 | —— |
| Network | 14.1 | 4.01 | 1.92 | —— | —— |
| GasSen | 783.78 | 450.34 | 350.52 | 148.89 | 145.759 |
| MSD | 80.47 | 64.44 | 44.9 | 43.65 | 41.44 |
| GasSenH | 16.59 | 13.35 | 11.76 | 11.04 | 10.16 |

Table 7: Average SPN sizes (# of nodes) as we vary the threshold for the maximum # of variables in a multivariate Gaussian leaf. No results are reported (dashes) when the maximum # of variables is greater than the total number of variables.

| Dataset | Maximum # of Variables per Leaf Node | | | | |
|---|---|---|---|---|---|
| | 1 | 2 | 3 | 4 | 5 |
| Power | 14269 | 2813 | 427 | 8 | —— |
| Network | 7214 | 1033 | 7 | —— | —— |
| GasSen | 13874 | 6879 | 5057 | 772 | 738 |
| MSD | 6547 | 3114 | 802 | 672 | 582 |
| GasSenH | 1901 | 1203 | 920 | 798 | 664 |

Table 8: Log Likelihoods for different correlation thresholds.

| Dataset | Correlation Threshold | | | | | |
|---|---|---|---|---|---|---|
| | 0.05 | 0.1 | 0.2 | 0.3 | 0.5 | 0.7 |
| Power | $-2.37 \pm 0.13$ | $-2.46 \pm 0.11$ | **$-2.20 \pm 0.18$** | $-3.02 \pm 0.24$ | $-4.65 \pm 0.11$ | $-4.68 \pm 0.09$ |
| Network | **$-3.98 \pm 0.09$** | $-4.27 \pm 0.02$ | $-4.75 \pm 0.02$ | $-4.75 \pm 0.02$ | $-4.75 \pm 0.02$ | $-4.75 \pm 0.02$ |
| GasSen | $-104 \pm 5$ | **$-102 \pm 4$** | **$-102 \pm 3$** | **$-102 \pm 3$** | $-103 \pm 3$ | $-110 \pm 3$ |
| MSD | **$-531.4 \pm 0.3$** | **$-531.4 \pm 0.3$** | **$-531.4 \pm 0.3$** | **$-531.4 \pm 0.3$** | $-532.0 \pm 0.3$ | $-536.2 \pm 0.1$ |
| GasSenH | **$-15.6 \pm 1.2$** | **$-15.6 \pm 1.2$** | $-15.8 \pm 1.1$ | $-16.2 \pm 1.4$ | $-16.1 \pm 1.4$ | $-17.2 \pm 1.4$ |

Table 9: Average times (seconds) as we vary the correlation threshold.

| Dataset | Correlation Threshold | | | | | |
|---|---|---|---|---|---|---|
| | 0.05 | 0.1 | 0.2 | 0.3 | 0.5 | 0.7 |
| Power | 197 | 183 | 130 | 39 | 10 | 9 |
| Network | 20 | 14 | 1.9 | 1.9 | 1.9 | 1.9 |
| GasSen | 370 | 351 | 349 | 366 | 423 | 142 |
| MSD | 44.3 | 43.7 | 44.3 | 44.0 | 43.0 | 30.3 |
| GasSenH | 11.8 | 11.7 | 11.9 | 13.0 | 12.0 | 15.1 |

To understand the impact that the maximum number of variables per leaf node has on the resulting SPN, we performed experiments where the minibatch size and correlation threshold were held constant for a given dataset while the maximum number of variables per leaf node varies. We report the log likelihood with standard error after ten-fold cross validation, as well as average size and average time in Tables 5, 6 and 7. As expected, the number of nodes in an SPN decreases as the leaf node cap increases, since there will be less branching. What's interesting is that depending on the type of correlations in the datasets, different sizes perform better or worse. For example in Power, we notice that univariate leaf nodes are the best, but in GasSenH, slightly larger leaf nodes tend to do well. We show that too many variables in a leaf node leads to worse performance and underfitting, and in some cases too few variables per leaf node leads to overfitting. These results show that in general, the largest decrease in size and time while maintaining good performance occurs with a maximum of 3 variables per leaf node. Therefore in practice, 3 variables per leaf node works well, except when there are only a few variables in the dataset, then 1 is a good choice.

Tables 8, 9 and 10 show respectively how the log-likelihood, time and size changes as we vary the correlation threshold from 0.05 to 0.7. A very small correlation threshold tends to detect spurious correlations and lead to overfitting while a large correlation threshold tends to miss some correlations and lead to underfitting. The results in Table 8 generally support this tendency subject to noise due to sample effects. Since the highest log-likelihood was achieved in three of the datasets with a correlation threshold of 0.1, this explains why we used 0.1 as the threshold in the previous experiments. Tables 9 and 10 also show that the average time and size of the resulting SPNs generally decrease (subject to noise) as the correlation threshold increases since fewer correlations tend to be detected.

Table 10: Average SPN sizes (# of nodes) as the correlation threshold changes.

| Dataset | Correlation Threshold | | | | | |
|---|---|---|---|---|---|---|
| | 0.05 | 0.1 | 0.2 | 0.3 | 0.5 | 0.7 |
| Power | 24914 | 23360 | 16006 | 2813 | 11 | 11 |
| Network | 11233 | 7214 | 9 | 9 | 9 | 9 |
| GasSen | 5315 | 5057 | 5041 | 5035 | 4581 | 490 |
| MSD | 672 | 672 | 674 | 674 | 660 | 448 |
| GasSenH | 920 | 920 | 887 | 877 | 1275 | 796 |

## 5 CONCLUSION AND FUTURE WORK

This paper describes a first online structure learning technique for Gaussian SPNs that does a single pass through the data. This allowed us to learn the structure of Gaussian SPNs in domains for which the state of the art was previously to generate a random network structure. This algorithm can also scale to large datasets efficiently.

In the future, this work could be extended in several directions. We are investigating the combination of our structure learning technique with other parameter learning methods. Currently, we are simply learning the parameters by keeping running statistics for the weights, mean vectors, and covariance matrices. It might be possible to improve the performance by using more sophisticated parameter learning algorithms. We would also like to extend the structure learning algorithm to discrete variables. Finally, we would like to look into ways to automatically control the complexity of the networks. For example, it would be useful to add a regularization mechanism to avoid possible overfitting.

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
