# Peer review of "Online Structure Learning for Sum-Product Networks with Gaussian Leaves"

_ICLR 2017 — rejected_

[Official Review · AnonReviewer5 · rating 4 · confidence 1 · 13 Dec 2016]
**A novel constructive algorithm for SPNs with Gaussian observations.**

The authors contribute an algorithm for building sum-product networks (SPNs) from data, assuming a Gaussian distribution for all dimensions of the observed data.  Due to the restricted structure of the SPN architecture, building a valid architecture that is tailored to a specific dataset is not an obvious exercise, and so structure-learning algorithms are employed.  For Gaussian distributed observations, the authors state that the previous state of the art is to chose a random SPN that satisfies the completeness and decomposibility constraints that SPNs must observe, and to then learn the parameters (as done in Jaini 2016).  In the contributed manuscript, the algorithm begins with a completely factorized model, and then by passing through the data, builds up more structure, while updating appropriate node statistics to maintain the validity of the SPN.

The above Jaini reference figures heavily into the reading of the paper because it is (to my limited knowledge) the previous work SOTA on SPNs applied to Gaussian distributed data, and also because the authors of the current manuscript compare performance to datasets studied in Jaini et al.  I personally was unfamiliar with most of these datasets, and so have no basis to judge loglikelihoods, given a particular model, as being either good or poor.  Nevertheless, the current manuscript reports results on these datasets that better (5 / 7) than other methods, such as SPNS (constructed randomly), Stacked Restricted Boltzmann Machines or Generative Moment Matching networks.

Overall: 
First let me say, I am not really qualified to make a decision on the acceptance or rejection of this manuscript (yet I am forced to make just such a choice) because I am not an expert in SPNs. I was also unfamiliar with the datasets, so I had no intuitive understanding of the algorithms performance, even when viewed as a black-box.  The algorithm is presented without theoretical inspiration or justification.  These latter are by no means a bad thing, but it again gives me little hold onto when evaluating the manuscript.  The manuscript is clearly written, and to my limited knowledge novel, and their algorithm does a good job (5/7) on selected datasets.  

My overall impression is that there isn't very much work here (e.g., much of the text is similar to Jaini, and most of the other experiments are repeated verbatim from Jaini), but again I may be missing something (this manuscript DOES mostly Jaini). I say this mostly because I am unfamiliar with the datasets.  Hopefully my reviewing peers will have enough background to know if the results are impressive or not, and my review should be weighted minimally.

Smallish Problems
I wanted to see nonuniform covariances in the data of the the toy task (Fig 3) for each gaussian component.

The SPN construction method has two obvious hyper parameters, it is important to see how those parameters affect the graph structure. (I submitted this as a pre-review question, to which the authors responded that they would look into this.)

[Official Review · AnonReviewer6 · rating 4 · confidence 2 · 19 Dec 2016]
**An algorithm to learn the structure of continuous SPNs in a single pass - too unfinished**

# Summary
This paper proposes an algorithm to learn the structure of continuous SPNs in a single pass through the data,
basically by "growing" the SPN when two variables are correlated.

## NOTE
I am not an expert on SPNs, and can not really judge how impressive the presented results are due to lack of familiarity with the datsets.

# Pro
- This looks like possibly impactful work, proposing a simple and elegant algorithm for learning SPN structure single-pass, rather than just using random structure which has been done in other work in the online settings.

# Con
- The paper is heavily updated between submission deadline and submission of reviews.
- The paper reads like a rush job, sloppily written - at least the first version.
- Comparison to literature is severely lacking; eg "several automated structure learning techniques have been proposed" followed by 6 citations but no discussion of any of them, which one is most related, which ideas carry over from the offline setting to this online setting, etc. Also since this work presents both joint structure & *parameter* learning, comparison to the online parameter learning papers (3 cited) would be appreciated, specifically since these prior approaches seem to be more principled with Bayesian Moment Matching in Jaini 2016 for example.
- I do not know enough about SPNs and the datasets to properly judge how strong the results are, but they seem to be a bit underwhelming on the large datasets wrt Random

# Remaining questions after the paper updates
- Table 3: Random structure as baseline ok, but how were the parameters here learned? Your simple running average or with more advanced methods?
- Table 1: you are presenting *positive* average log-likelihood values? This should be an average of log(p<=1) < 0 values? What am I missing here?

I recommend reject mostly because this paper should have been finished and polished at submission time, not at review deadline time.

[Official Review · AnonReviewer4 · rating 6 · confidence 3 · 20 Dec 2016]

The authors present an online learning method for learning the structure of sum-product networks. The algorithm assumes Gaussian coordinate-wise marginal distributions, and learns both parameters and structure online. The parameters are updated by a recursive procedure that reweights nodes in the network that most contribute to the likelihood of the current data point. The structure learning is done by either merging independent product Gaussian nodes into multivariate leaf nodes, or creating a mixture over the two nodes when the multivariate would be too large.

The fact that the dataset is scaled to some larger datasets (in terms of the number of datapoints) is promising, although the number of variables is still quite small. Current benchmarks for tractable continuous density modeling with neural networks include the NICE and Real-NVP families of models, which can be scaled to both large numbers of datapoints and variables. Intractable methods like GAN, GenMMN, VAE have the same property. 

The main issue with this work for the ICLR audience is the use of mainly a set of SPN-specific datasets that are not used in the deep learning generative modeling literature. The use of GenMMN as a baseline is also not a good choice to bridge the gap to the neural community, as its Parzen-window based likelihood evaluation is not really meaningful. Better ways to evaluate the likelihood through annealed importance sampling are discussed in "On the Quantitative Analysis of Decoder-Based Generative Models" by Wu et al. I would recommend the use of a simple VAE type model to get a lower bound on the likelihood, or something like Real-NVP.

Most neural network density models are scalable to large numbers of observations as well as instances, and it is not clear that this method scales well "horizontally" like this. Evaluating the feasibility of modeling something like MNIST would be interesting.

SPNs have the strength that not only marginal but also various type of conditional queries are tractable, but performance on this is not evaluated or compared. One interesting application could be in imputation of unknown pixels or color channels in images, for which there is not currently a high-performing tractable model.

Despite the disconnect from other ICLR generative modeling literature, the algorithm here seems simple and intuitive and convincingly works better than the previous state of the art for online SPN structure learning. I think VAE is a much better baseline for continuous data than GenMMN when attempting to compare to neural network approaches. Further, the sum-product network could actually be combined with such deep latent variable models as an observation model or posterior, which could be a very powerful combination. 

I would like it if these SPN models were better known by the ICLR probabilistic modeling community, but I do not know if this paper does enough to make them relevant. As with the other reviewers, I am not an expert on SPNs. However, this seems to be a simple and effective algorithm for online structure induction, and the scalability aspect is something that is important in much recent work in the learning of representations. I think it is good enough for publication, although I would prefer to see many of the above additions to more clearly bridge the gap with other literature in deep generative modeling.

[Public Comment · Sang-Woo Lee · 04 Jan 2017]
**Comparing to the paper "Online Incremental Structure Learning for Sum-Product Networks" (ICONIP, 2013)**

Hi. I hope the paper mentions my previous work "Online Incremental Structure Learning for Sum-Product Networks" (ICONIP, 2013).
I think the idea is quite similar, both for online parameter learning and online structure learning, although there is no consideration over Gaussian leaves in my paper.

[Final Decision · Program Chairs · 06 Feb 2017]
**ICLR committee final decision**

The authors present a framework for online structure learning of sum-product network. They overcome challenges such as being able to learn a valid sum product network and to have an online learning mechanism. Based on the extensive discussions presented by the reviewers, our recommendation is to accept this paper for a workshop.